# Horse Behavior, Physiology and Emotions during Habituation to a Treadmill

**DOI:** 10.3390/ani10060921

**Published:** 2020-05-26

**Authors:** Malgorzata Masko, Malgorzata Domino, Dorota Lewczuk, Tomasz Jasinski, Zdzislaw Gajewski

**Affiliations:** 1Department of Animal Breeding, Institute of Animal Science, Warsaw University of Life Sciences (WULS—SGGW), 02-787 Warsaw, Poland; malgorzata.masko@sggw.edu.pl; 2Department of Large Animal Diseases and Clinic, Veterinary Research Centre and Center for Biomedical Research, Institute of Veterinary Medicine, Warsaw University of Life Sciences (WULS—SGGW), 02-787 Warsaw, Poland; tomasz_jasinski@sggw.edu.pl (T.J.); zdzislaw_gajewski@sggw.edu.pl (Z.G.); 3Institute of Genetics and Animal Breeding, Polish Academy of Sciences Jastrzębiec (PAS—PAN), 05-552 Magdalenka, Poland; d.lewczuk@ighz.pl

**Keywords:** habituation, ethogram, test, behavior, horses

## Abstract

**Simple Summary:**

Treadmills have become a popular and important tool in various aspects of equestrianism. Habituation to treadmill locomotion has been investigated mainly in the biomechanic aspect. The behavioral aspect of habituation was seldom described; therefore, it was not clear if the habituation process was the same in horses with different temperaments and emotional responses to the handler. We assessed the results horses got in the novel object test, the handling test, and both positive and negative emotional response tests, and their connection with behavior-related features of the habituation process. The four principal components in the examined horses were identified: “Flightiness,” “Freeziness,” “Curiosity,” and “Timidity”. We found that the features of Flightiness gradually decreased during habituation. A part of them increased again when the horses started a new challenge. Features of Freeziness and Curiosity showed strong stability throughout the whole habituation. Features of Timidity strongly increased when the treadmill was introduced; thus, the challenge was completely changed. The first entrance and work on a treadmill seemed to cause fright responses. We conclude that the habituation process should be adapted to the horse’s temperament and emotionality. Such findings will improve handler safety and lead to increased horse welfare during habituation.

**Abstract:**

A treadmill is an important tool in the equine analysis of gait, lameness, and hoof balance, as well as for the evaluation of horse rehabilitation or poor performance including dynamic endoscopy. Before all of these uses, horses have to be habituated to a treadmill locomotion. We used principal component analysis to evaluate the relationship between aspects of the horse’s temperament and emotional response, and progress in the behavioral habituation to a treadmill. Fourteen horses were tested, by the same familiar handler, using the novel object test, the handling test, and both positive and negative emotional response tests. Then, four stages of gradual habituation of the first work on a treadmill were conducted. Each time, the horse’s behavior was filmed. Data obtained from ethograms and heart rate measurements were tested. Four principal components were identified in examined horses: “Flightiness”, “Freeziness”, “Curiosity”, and “Timidity”. Flightiness was connected with nervousness, agitation by new objects, and easy excitability, and gradually decreased of features during habituation. Timidity was associated with a lack of courage and stress in new situations, and those features strongly increased when the treadmill was introduced. Freeziness and Curiosity features showed strong stability throughout the whole habituation. The results of this study provide evidence for a connection between temperament, emotional response, and habituation process in a horse.

## 1. Introduction

Horses’ behavioral problems can manifest themselves during housing, handling, riding, and/or working from the ground. Horses exposed to inadequate housing have a high risk of developing stereotypies [1]. The stereotypies develop more often in the cases of restricted space, the lack of movement, and the housing of horses in isolated, individual conditions [1,2,3]. Similarly, transport-related problem behaviors may also develop when people do not train their horses for loading and travelling in transport vehicles, or when they use inappropriate training methods [4,5,6]. Such problems may also apply to horses getting used to working on a treadmill. Horse treadmills operate much like human treadmills but on a much larger scale. As opposed to the round pen or racetrack work, the horse is moving forward on a cushioned moving belt at a known speed. Treadmill work takes out variables such as weather, rider error, footing issues, or inconsistent pace [7,8,9,10,11,12,13]. The treadmill is commonly used to investigate poor performance problems and perform upper airway dynamic endoscopies [7]. It is also an important tool in equine gait analysis [8], lameness [9], and hoof balance [10] evaluation or horses rehabilitation [11]. The use of the treadmill allows for measurements of the horse’s physiology and biomechanics, allowing researchers to obtain valuable data equivalent to overground locomotion. However, before a reasonable comparison can be made, horses have to be habituated to treadmill locomotion [12]. Habituation to treadmill locomotion has been investigated mainly in the biomechanic aspect, whereas behavioral observations are seldom described [9,12,13]. Due to the fact that inappropriate techniques of handling and/or working from the ground account for much of the wastage rates among horses, and also the majority of injuries among handlers [14], the association between horses’ temperament and behavioral habituation of horses to the work on a treadmill are an important aspect of the equine industry.

The first work on a treadmill may cause fright responses or evasive behavior [13,15]. Evasive behavior results from a conflict situation in the horse and is usually characterized by hyperreactivity and arises largely through confusion [2]. Rearing, throwing themselves backward or sideways, and kicking occurs because the escape is suspected as a horses’ first reaction to frightening stimuli [16]. Both rearing and throwing behaviors may be amplified when the handling staff does not oppose these behaviors [17]. Horses tend to respond with avoidance or flight to unfamiliar situations or potential dangers. With repeated exposure, however, horses become accustomed to their surroundings and cease to avoid nonthreatening stimuli. This waning of responsiveness towards a repeated stimulus is termed “habituation” [16]. During the behavioral habituation to the work on the treadmill, the handler has to respond carefully and interplay with the horse’s behavior and on the horse’s behavioral reactions toward new situations. The handler should know the techniques for behavior modification of fear reactions include desensitization, which involves gradually exposing the animal to the eliciting stimulus without provoking the undesired response [16]. The most commonly used method for horse learning is based on positive reinforcements [2]. The training based on negative reinforcement and positive punishment is not consistent with best practice for teaching horses for example to load and travel [4,5]. During habituation, there are two valuable and complementary indicators of horse’s motivations and fear levels. First are the leading rates, which provide information about the horse reactions to the start and stop signals [16], and the second are the stress rates [18]. Leading rates include behavioral observations and measurement of the working time, which increases when the horse manifests a reluctance to follow a human [19]. Stress rates estimate the physiological reaction based on behavioral observations and heart rate measurements. The normal heart frequency is about 44 bpm (beats per minute), and range 23 to 70 bpm [18]. Heart rate increase over the norm as a result of both intensified physical activity and/or preparedness for new situations: flying [20,21] or transporting [6].

The interaction and communication between horse and handler depend on the skills of the handler and the horse trust based on earlier experience. The proper use of the learning theory is then crucial for effective habituation [2], the correct timing, consistent responses and signals influence strongly on the final effect of horses’ work on the treadmill [9]. However, this is only a part of the training’s complexity. The relationship between horse and handler is interactive, complex, and two-way [22]. Therefore, another factor is the horse’s temperament. A horse’s temperament has been assessed both qualitatively and quantitatively by using rating scores from assessors and by using behavioral tests [23,24]. The behavioral tests can be carried out in a more standardized way compared with the assessment made subjectively. These tests are being used to show horse responsiveness and motivations, especially emotional reactivity, reactions to human handling, and learning abilities [25,26]. Therefore, evaluating the interaction between the horse’s temperament, emotional response, and behavioral habituation remains a very subjective task.

The objective of this study was to determine the feasibility of equine habituation evaluation processes to the treadmill with various behavior-related features measured before the habituation process. The strong connection is expected between investigated horse behavior traits as well as the emotions between horse and human. To test this hypothesis, we have adapted ethograms and behavior analyses usually used in the evaluation of the horse’s temperament, emotional response, and habituation process. We have used principal component analysis to evaluate the relationship between aspects of the horse’s temperament and emotional response, and progress in the behavioral habituation to the work on the treadmill.

## 2. Materials and Methods

Fourteen horses participated in the study (six mares and eight geldings, mean age 14.1 ± 4.68 years, mean weight 546 ± 47.8 kg) of Polish warmblood breeds. All horses were housed in individual stalls with the same management in the Didactic Stable of Horse Breeding Division at WULS (Warsaw University of Life Sciences). The horses were fed three times a day with a dose of oats and hay personalized to each horse to maintain an optimal, healthy condition without obesity, and had daily access to a sandy paddock no shorter than 6 h per day. All horses were in daily leisure use, namely recreational riding 1 to 2 h a day, five days a week. All horses had never been worked or entered on any treadmill before. Animal care and experimental procedures were in accordance with the guidelines for the treatment of animals in behavioral research and teachings by the Association for the Study of Animal Behavior [27]. No evidence of the horses’ social frustration was found throughout the experiment. All procedures, behavioral tests, and habituation took place in a familiar, daily used environment and did not cause any pain, suffering, or damage to the horses.

All procedures lasted 15 weeks and included temperament tests, emotional response tests, and the habituation process. In all tests, each horse was led by the same familiar handler. The behavioral tests, including temperament tests and emotional response tests, were carried out in the following order: the novel object test, the negative emotional response test, the handling test, and the positive emotional test. The habituation of the first work on a treadmill lasted 10 weeks and included four-stages of gradual habituation. Each stage was repeated three times, excluding the first-stage test and the final work test, which were repeated once. The interval between tests or repetitions during the habituation process lasted always seven days. The subsequent experiment procedures are marked on the timeline in Figure 1. 

For each stage, another structure from obstacle elements, named imitation of the treadmill, was used. The imitation of the treadmill was built from wood elements, poles, and standers, in order to achieve an appearance similar to the treadmill. Horses were used to such obstacle elements. The wall-filling increased gradually and the platform was attached. Finally, the test session on the real treadmill was performed. The imitation of the treadmill and then the treadmill was situated in the middle of the ridding hall well known for all horses. In the first-stage test, horses walked through the corridor built of two 3-m poles suspended on stands at height 0.60 m and 1.20 m. The distance between the walls of the corridor was 1.20 m (Figure 1D). In the second-stage test, the corridor was built using six 3-m poles suspended on stands at height 0.60 m and 1.20 m. The distance between the walls remained unchanged, but the ground was filled by a rubber mat: width 1.20 m, length 3.00 m (Figure 1E). In the third-stage test, the walls of the corridor were the same as in the second one; however, the ground was filled by special platform: height 0.45 m, width 1.20 m, length 3.00 m, with the ramp: length 1.00 m (Figure 1F). In the fourth-stage test, the test horse entered for the first time into the real treadmill. Horses stood three minutes inside and went down without any work (Figure 1G). The treadmill with transparent sidewalls and the following inside dimensions: height 1.95 m (including platform height 0.45 m), width 1.20 m, the length between the bars in front and back 4.20 m, and total length 6.80 m (including two ramps each with a length 1.30 m) was used (Master-Sport treadmill, Skarzysko-Kamienna, Poland). In the test session, horses started with the walk on the treadmill with a speed up to 1.6 m/sec. The test session lasted for 5 min.

### 2.1. The Horse’s Temperament Evaluation

The tests for temperament were adapted from a number of tests that have been used in other studies [25,28]. Horses were tested for temperament using a novel object test and a handling test.

During the novel object test, a familiar handler led the horse into a familiar indoor arena and left the horse in a so-called starting box in a corner of the arena. After being left to settle down for approximately 2 min, the horse was released out of the starting box through an automatic sliding door and was free to move in the indoor arena. After 2 min, an open blue and white umbrella was lowered from the ceiling. The horse’s behavior was videotaped during the time 2 min before and 5 min after exposure to the umbrella. Thereafter, the horse was caught and brought back to the stable. 

During the handling test, the horse was led by a familiar handler into the familiar indoor arena where the horse was encouraged to follow the handler walking over the bridge lying on the ground and made by plywood plates: width 2.00 m, length 4.00 m. Horses were not forced to follow; however, the handler kept a slight tension on the rope in order to guide the horse to go forward. When horses reared, shied, or went backward, another attempt was made. Each horse was allowed to make a maximum of three attempts and then was brought back to the stable. The handling test duration lasted between 2 and 5 min when the horse’s behavior was filmed.

### 2.2. The Horse’s Emotional Response Evaluation

The test for the emotional response was adapted from the previous studies. Horses were tested for emotional response twice, the first time using the negative test and the second time using the positive test [29]. 

During the emotional response test, a familiar handler led the horse into the middle of a round familiar indoor arena, diameter: 18 m, and removed the halter when both were facing the novel object. The novel object was the plastic bin covered with blue and white or blue and yellow shower curtain. The horse was free to move in the indoor arena beginning from the start location at approximately 7 m distance from the subject and handler. The handler alternated gaze and voice between the novel object and the horse. In the positive test, the positive emotional expression was transferred using excited facial expression, relaxed body posture and positive vocalization: “great” repeated every 10 s. In the negative test, the negative emotional expression was transferred using anxious facial expression, tense body posture, and negative vocalization “stop” repeated every 10 s. The emotional response test duration lasted between 1 and 2 min when the horse’s behavior was videotaped. Thereafter, the horse was caught and brought back to the stable.

### 2.3. The Horse’s Habituation Evaluation

For the habituation, the horse was equipped with the heart rate measuring device (Polar Equine, Healthcheck FT1 NC) in their stables. The HR-receiver was placed on the left side of the horse’s chest, just behind the front leg. During all steps of habituation and the test session, after a resting heart rate of 2 min was taken, a familiar handler led the horse into a familiar indoor arena. On the arena where the horse was encouraged to follow the handler walking over one of the treadmill imitations or the real treadmill. Each work always started at the same site, 5 m from the front of the treadmill imitation or treadmill. The horse was led forward from left to right in the walk. Horses were not forced to follow; however, the handler kept a slight tension on the rope in order to guide the horse to go forward. The positive reinforcement included a piece of carrot or apple with the voice command “great” after each series of steps forward. The positive punishment included a strengthening of the pressure from the rope with the “stop” voice command when the horse manifested any reluctance behavior while approaching the treadmill imitation or treadmill, such as rearing, walking backwards, walking sideways, kicking, or head shaking. The positive reinforcement and the positive punishment were used during the habituation process immediately after desired or unwanted behaviors respectively. All horses were conditioned in the same way. The work finished when all four feet were on the treadmill imitation or treadmill. The horses stood inside for 3 min with the positive reinforcement and thereafter the horse was led forward outside. Each work duration lasted between 2 and 10 min when the horse’s behavior was videotaped. Thereafter, the horse was brought back to the stable.

### 2.4. Data Collection and Analysis

Each test was recorded by a wide-angle camera (GoPro, Hero 3), placed on the edge of the indoor arena in such a way that all horses were visible without any restriction all the time. Videotapes were analyzed using the technical software (Observer, v 4.1). The frequency or duration of consecutive behaviors was measured using ethograms described for the evaluation of a horse’s temperament in Table 1, a horse’s emotional response in Table 2, and a horse’s habituation in Table 3. The duration of time-depended behaviors was calculated as a percentage of the total test time. Heart rate (bpm) for each horse was noted three times during each repetition of the stages of the habituation and the test session on a treadmill. The first time, immediately before the work in front of the imitation of the treadmill or the real treadmill. The second time, inside the treadmill imitation or treadmill when all four hoofs were set down. Lastly, the third time, 30 s after the horse left the imitation of the treadmill or treadmill and was stopped behind.

Obtained data are represented the form of matrix X∈ℝt×n with elements xj,i, where j index correspond to realizations and i index corresponds to features being univariate marginals. Each test day we have received t = 14 realizations representing subsequent examined horses. In our approach we took under consideration 37 features: nine features in evaluating a horse’s temperament (SN, FNO, HL, LTNO, TC, TU, NT, RB, SSB), seven features, measured twice, in evaluating a horse’s emotional response (negative GH, GO, IH, IO, BEx, AEx, IFEx; positive GH, GO, IH, IO, BEx, AEx, IFEx), and 14 features in evaluation of horse’s habituation (SN, FNO, RB, SSO, WH, DF, SH, CH, ST, TUC, FN, HR I, HR II, HR III), of which the order is consistent with indexing. For each day of the habituation process, comprising a total of 11 repetitions, we have collected data in the following matrix form: X∈ℝ14x14. In our model, we obtained 11 matrices X∈ℝt×n, representing the subsequent days of habituation: the first-stage test, the second-stage test (1st, 2nd, 3rd reps), the third-stage test (1st, 2nd, 3rd reps), the fourth-stage test (1st, 2nd, 3rd reps), and the test session. For each of the received matrix, the single realization vector was xj=xj,1,…,xj,n∈ℝ1×37. 

The relationships between horse temperament and emotional response, assessed in the behavioral tests, as well as horse’s habituation process to the work on the treadmill, assessed in the behavioral observation and the heart rate measurements, were tested using principal component analysis (PCA) and factor analysis (FA). PCA was employed to summarize correlated measures into so-called principal components [25,32]. PCA are linear combinations of the original variables and represent condensed new variables reflecting independent characteristics underlying the correlation matrix. The loading of each measure on a principal component represents the correlation between the new variable and the original measure and on this basis indicates the importance of a measure for a principal component [25]. The covariance matrix was applied as all features used the same scale (whole numbers of 1–9), for this purpose, each j for each i was scaled. Principal component scores were retained for interpretation and analysis if the eigenvalue was >1.0, if percent variance explained was $>\frac{1}{i}$, where i is the number of measured features (1/37 = 2.70%), and by examining the screen plot for the point where the eigenvalue slope plateaued. For all features, behaviors loading on the positive side of the FA have a score closer to 9, indicating that the horses received the maximum value of the examined feature. Behaviors loading on the negative side of the FA have a scores closer to 1 and received the minimum value of the examined feature. The first four principal components (PCs) were interpreted as meaningful behavior data based on the percentage variance of each feature obtained for the evaluation of horse’s temperament, emotional response, and habituation. The general model included average values from data for each day of habituation. 

The features measured during the equine habituation process were compared between subsequent habituation days: the first-stage test, three repetitions of the second-stage test, three repetitions of the third-stage test, three repetitions of the fourth-stage test, and the test session. Since these features were not normally distributed, even after applying appropriate scaling, a non-parametric statistical test was used. Since each represents a single horse, each row represents matched measures; therefore, Friedman test was chosen. Dunn’s post hoc test for multiple comparisons followed each Friedman test with a significant result. Statistical analysis was performed using GraphPad Prism6 software (GraphPad Prism 6; GraphPad Software Inc., San Diego, CA, USA), where the significance level was established as *p* < 0.05.

## 3. Results

The overall goal of this study was to determine the feasibility of evaluating equine habituation processes with diverse behavior-related features. To test this hypothesis, we performed a statistical analysis of behavioral features obtained in the study. These features, used in the evaluation of the horse’s temperament, emotional response, and habituation process, were transformed to 1–9 scores, and then, profiles for four of the behavioral factors were performed using PCA, to identify the importance of temperament and emotional response during habituation.

### 3.1. The Factors Explaining Behavioral Variation

We conducted PCA and FA to explore the degree of behavioral features variation during the whole habituation process, including to the general model average values from 11 repetitions conducted during habituation. Principal components 1 through 4 cumulatively accounted for 67% of the total variance (Figure 2A). All features were labeled as belonging to a factor, based on behaviors with absolute loading values more than 1.0 (Figure 2B). 

The first PCA (Factor 1) was labeled “Flightiness.” The percentage of total variation explained by this component was 35%. The following features were subjected to PCA “Flightiness”: Focused on the novel object during the evaluation of horse’s temperament (FNO ^a^); Head low (HL); Latency time (LTNO); Trot and canter (TC); Tail up (TU); Abreast to handler during negative test (nAEx); Focused on the novel object during the evaluation of horse’s habituation (FNO ^b^); Defecating (DF); Stepping (ST); Flared nostril (FN); and Heart rate before work (HR I). This component had high positive loadings for both FNO ^a^, TC, and FN (Figure 2C).

The second PCA (Factor 2) was labeled “Freeziness”, and included: Total needs (NT); Reluctance behavior observed during the evaluation of horse’s temperament (RB ^a^) and habituation (RB ^b^); Standing still bridge (SSB); Standing still obstacle (SSO) and Whinnying (WH). The percentage of total variation explained by this component was 45%. This component had high positive loadings for NT, RB ^b^ and WH (Figure 2C).

The third PCA (Factor 3) labeled as “Curiosity” explained 58% of the total variation. PCA “Curiosity” included the following features: Snorting observed during the evaluation of horse’s temperament (SN ^a^) and habituation (SN ^b^); Tail up or croved (TUC); as well as Gaze object (nGO; pGO), Interaction object (nIO; pIO) and In front of handler (nIFEx; pIFEx) observed during negative and positive test, respectively. “Curiosity” had high positive loadings for nGO; pGO; nIO; pIO; nIFEx; pIFEx; and SN ^b^ (Figure 2C).

The last fourth PCA (Factor 4) was labeled “Timidity” and explained 67% of the total variation. This PCA included the following features: Shaking (SH); Chewing (CH); Gaze human (nGH; pGH), Interaction human (nIH; pIH) and Behind handler (nBEx; pBEx) observed during negative and positive test, respectively; and Heart rate during work (HR II) and after work (HR III). This component had high positive loadings for both GH; IH and BEx; for CH and two HRs (Figure 2C).

### 3.2. The Changes in Behavioral Features during Habituation

In the first PCA, labeled “Flightiness,” scaled values of FNO ^b^; DF; ST and HR I were compared, and the significant differences between the consecutive days of habituation process were demonstrated (Figure 3). Dunn’s test indicated that FNO ^b^ decreased gradually (*p* = 0.024) from the first-stage test and was significantly lower in the 3rd repetition of the third-stage test. When the type of object was completely changed, from obstacles to a treadmill, FNO ^b^ increase to the level measured at the beginning of habituation (Figure 3A). Similarly, the values of Flared nostril decreased gradually from the first-stage test to the 3rd repetition of the third-stage test (*p* < 0.0001), and then increased when the new object was introduced however to the higher level than at the beginning of habituation (Figure 3D). The values of Stepping (*p* = 0.018; Figure 3C) and Heart rate before work (*p* = 0.032; Figure 3E) also increased significantly when the new object was introduced. The gradual decrease in values of ST and HR I was noted throughout the repetitions of the third-stage test. The changes were more pronounced in ST than in HR I. When the horses started a new challenge, the test session, a significantly increase of both values were observed again. Each introduction of a new challenge (changed the stage of the tests) resulted in a significant increase of Defecating (*p* = 0.013). The values of DF gradually decreased from 1st to 3rd repetition of each test; however, the values after the introduction of a new object, treadmill, were the highest. Additionally, after starting work on the treadmill, DF increased again (Figure 3B).

The comparison between the consecutive days of the habituation process for the features of the second PCA, labeled “Freeziness”, indicated significant differences for SSO but not for RB and WH (Figure 4). The values of the Standing still obstacle increased significantly (*p* < 0.0001) when the new challenge was introduced; however, no rule was found in the change in value with subsequent repetitions. In the second-stage test, SSO was higher in 1st than in 2nd and 3rd repetitions. In the third-stage test, SSO was higher in all three repetitions than in the first-stage test, whereas in the fourth-stage test SSO was higher in the 1st and 2nd than in 3rd repetition. There was also no increase in SSO during the test session (Figure 4B).

In the third PCA, labeled “Curiosity,” scaled values of SN and TUC were compared, and no significant differences between the consecutive days of habituation process were noted (Figure 5).

In line with features of “Flightiness,” the features of the fourth PCA, labeled “Timidity,” significantly differed between the consecutive days of habituation process (Figure 6). Scaled values of SH; CH; and HR II and HR III were in general higher when the new object, treadmill, was introduced. The values of Shaking were higher (*p* = 0.006) during each work on the treadmill than during works with obstacles made by the familiar poles and standers (Figure 6A). Similarly, the values of Chewing were higher (*p* = 0.005) during each work on a treadmill. However, during works with treadmill imitation, the values of CH decreased with subsequent test repetitions (Figure 6B). Heart rate during work was higher (*p* < 0.0001) during the 1st and 2nd repetitions of test on a treadmill and during the test session on a treadmill than in the other tests (Figure 6C). Interestingly, heart rate after work was significantly higher (*p* < 0.0001) only in the 1st repetition of the test introducing novel object, compared to the second-stage test and the fourth stage test (Figure 6D).

## 4. Discussion

Behavior and temperament assessments are extremely important aspects considered as a key issue in horse health and performance [33], and probably in habituation to the novel type of work as well. Understanding the relationship between the habituation process and the temperament and emotional responses of horses can lead to improved selection for appropriate tasks, and therefore lead to improved management and welfare for the horse, as well as improved safety for their handlers. In this study, we identified four behavioral components and have successfully described the changes of behavioral features typical for these components during habituation to the first work on a treadmill.

Currently, there are three types of behavioral assessments in horses, the subjective assessments on the point scale with or without questionnaire survey completed by familiar handlers or judges [34,35,36], and the behavioral tests [6,25,31,36,37]. The first reflects a respondent’s personal impressions and experiences and is thus subjective. On the other hand, a questionnaire is based on long-term observation and could include several innate emotional traits that are stable across time [34,38]. The second one, the behavioral test, is more objective; however, it reflects a ‘snapshot’ in time without any emotional relations between the horse and the handler [38]. Therefore, we adopted the typical behavioral test to include the exchanging of emotional information between horse and human. The handlers’ emotional expression influences the behavior of horses in novel situations and might be of interest for training and husbandry practices [29]. We suspected that this knowledge is of great interest to the habituation process. The classical leading test, just like the Handling Test [25], provides information about the horse’s reactions to the start and stop signals [7,16] and a readiness or reluctance to follow a human [19]. However, in this test it is only the familiar handler who leads a horse, without any information about the handler emotional expression and knowledge of the horse, assumed to be neutral. It should be noted that the most commonly method used for horse learning and habituating is based on reinforcements [2,19]. The use of training based on negative reinforcement and positive punishment is not consistent with best practice for the equine habituation, for example when teaching horses to load and travel [4,5]. Moreover, the horse’s readiness to respond to these reinforcements should be included in the evaluating of the equine habituation process. We also conducted the classic novel object test, where a novel stimulus elicits behaviors related to fear and a startle response through the innate startle reflex [39]. In this test, horses, without any impact of a handler, demonstrate a natural fear of new surprising objects [39] without any impact of a handler. The fear of novel objects corresponds closely with the fear of unknown restricted spaces. Horses, as an innately neophobic animal, demonstrate a fear of restricted spaces, namely the tight, narrow space of a trailer [21] as well as the narrow, cage-shaped inside of a treadmill [13]. Therefore, the tendency of horses to manifest a natural fear reaction should be taken into account when the habituation process is evaluated. Ethograms of behaviors analyzed with the Novel Object Test and the Handling Test used by Visser et al. (2008) and the ethogram of horse’s emotional response developed by Schrimpf et al. (2020) had already proven successful tools for the study of equine temperament [25,29]. Similarly, our ethogram used in the evaluation of the habituation process is based on the Yngvesson et al. (2016), Stomp et al. (2018), and Padalino et al. (2018) studies, and thus, seems to be adequate to provide preliminary insights to behaviors variation in horses [6,30,31].

Principal component analysis and factor analysis of behavior feature from temperament, emotional response and habituation tests show fairly consistent trends in flightiness, freeziness, curiosity, and timidity profiles. It may be suggested that we observed the individual differences in the horses’ flight response. The flight response should be understood as an adaptation that has been crucial for horses’ survival during evolution, which is not subject to fundamental changes by domestication but only is subject to some individual fluctuations [40]. These individual differences observed in the adult horses may also result from early experiences. Young age is an important time period for behavioral development. Different experience factors can influence this development, including housing, social life, and human-foal interaction [41]. In our studies, horses with high scores on the Flightiness principal component exhibited tend to become nervous or agitated by new objects and tend to get excited more easily or respond too quickly to new stimuli. They also exhibited high average heart rates before work but not during and after it. The heart rate parameters were widely used in measuring mental stress in horses [26,42]. The flightiness horses, showed signs of mental stress, anxiety, and nervousness mainly before starting the work with a new object. Similar to the recent research [32,38,43], our studded features related to anxiety and nervousness consistently account for the greatest amount of variation, taking into account the heart rate parameters as indicators of mental stress in horses [23,42]. It should be noted that more severe signs of stress during and after work were observed in timid horses. Horses with high scores on the Timidity principal component exhibited trends to be timid or lack courage in novel environments and tend to get stressed in new situations without expression of any other stress behaviors. These horses seem to be strongly emotionally connected with a handler or emotionally dependent on the handler, and regardless of the human negative or positive emotional reinforcement, they behaved like horses under the influence of the positive punishment. Timid horses indicated a higher vigilance or alertness like horses receiving a negative emotional expression [29], which may be explained by horses’ species-typical behavior during potential threats. Naive horses show strong fight-or-flight response as a sign of preparation [40].

Our results indicate that these horses did not show reluctance behavior, despite fear and uncertainty, when performing the challenge in a novel situation. Reluctance behavior and a prolonged time for standing still in front of the bridge or structure of obstacles were typical for the horses with high Freeziness scores. Horses with high scores on this exhibited principle component tend to be obstinate once it resists a command and tend to be patient with various stimuli. Our results are in contrast to previous research, indicating that more reluctance behavior during work was shown to be more emotionally reactive horses [25]. In our research, the freezing horses exhibited poor emotional reactions and also poor responses to various stimuli. Similar perseverance was demonstrated by the curious horses. Horses with high scores on the Curiosity principal component tend to be interested in novel objects, approach them, and tend to be cooperative and willing to work with the handler. These horses seem to be strongly emotionally independent of a handler and exhibited signs of good mood when taking up work with a new object. Following Stomp et al. (2018), we considered “snorting” as an indicator of a relaxation phase associated with positive emotions [30]. Minero et al. (2018) found that more relaxed horses show less avoidance behavior towards humans [44]. We even suppose that more relaxed and inquisitive horses are more interested in novel objects, regardless of the human negative or positive emotional reinforcement. 

Habituation to treadmill locomotion was investigated mainly in the biomechanic aspect, whereas behavioral observations are seldom described. The pattern of horses walking and trotting on a treadmill during a training period of one week [12], kinetic data over 10 training sessions [9], the time required to obtain steady-state locomotion [12,13], and the minimal number of training sessions needed to allow reliable numerical gait analysis [9] were previously discussed. In a behavioral aspect, habituation to the first work on a treadmill is a serious challenge, similar to the loading a horse onto a trailer [2]. Naive horses are stressed by being loaded onto the trailer [16,45], whereas the behavioral signs of stress decrease with increasing transport vehicle size [17]. A previous study demonstrated that habituation can reduce this stress [6,46], which allows the horse to generalize the new stimuli occurring during the work with new objects, and react more calmly [20]. Our results suggested that this generalization of new stimuli may differs depending on the general horse’s temperament and tendencies in emotional response. Thus, the handler’s responses to the horse’s behavioral reactions toward habituation should interplay with the horse’s temperament. Especially since Schrimpf et al. (2020) has demonstrated that horses can adjust their behavior towards the environment by taking into account human emotions [30]. Due to certain limitations of the experimental design, we may only suggest the temperament-depend habituation. By using only one, small experimental group, we cannot divide subsequent horses to the best matched principal components group and then compared the habituation process in each PC. More research is needed to confirm the attachment our observations as a typical for Flightiness, Freeziness, Curiosity, or Timidity.

The values of features of Flightiness PC gradually decreased during habituation. A part of them increased again when the type of object was completely changed or when the horses started a new challenge, the test session. This is typical for the classic habituation process, when horses learn to limit their emotional response, and with each subsequent repetition react more calmly [6,20]. For these features, and for these type of horses, the classic handling methods during habituation are recommended. The values of features of Freeziness and Curiosity PCs shown strong stability throughout the whole habituation process, with a lack of typical differences between stages of repetitions. For the discussed features, the classic handling methods during habituation are also recommended. Finally, the values of features of the Timidity PC strongly increased when the treadmill was introduced; thus, the required challenge completely change. The first entrance and work on a treadmill seem to have caused fright responses, similar to what was noted in recent studies [13,15]. Fright response results arise from a conflict situation in the horse and are usually characterized by hyperreactivity and arise largely through confusion [2]. The timid horses did not show any reluctance behavior, such as rearing, throwing themselves backward or sideways, and/or kicking, which should occur because the escape is suspected as a horses’ first reaction to frightening stimuli [16]. Perhaps these horses interpret subtle human head and body cues and showed differing behavior depending on the handles’ signals and emotions [47,48]. However, due to the lack of clear behaviors typical of aversion, stress, and resistance, the possibility of handler reaction is limited. During the classic habituation to the work on the treadmill, the handler has to respond and interplay with the horse’s behavior and on the horse’s behavioral reactions toward new situations. The classic habituation is one approach to desensitizing horses to their innate fear of a new situation [49]. The technique commences with a series of simulations of activities performed around, in, and/or with the new stimulus until the horse no longer shows anxiety [4]. To achieve this goal, methods based on positive reinforcements [2,19] and positive punishment [4,5] were used, though always with concern to the direct interaction between human and horse [6]. In the case of Timidity, it may be difficult; thus, special handling methods during habituation, so-called alternative habituation, are recommended. Self-loading, as a possibility of horses training to load freely to the transport vehicles [17], and shaping of alternative responses, e.g., encouraging the animal to engage in a competing behavior that is incompatible with the undesired response [50], could be examples of the alternative habituation. However, the handling strategies in connection to horse personalities require explanation. Further research is warranted to find out the special handling methods in connection to horse personalities, because both stress and inexpedient reactions may be amplified when the handling staff does not react accordingly [17], even if typical behaviors are not manifested by the timid horses.

## 5. Conclusions

The results of this study provide evidence for a connection between temperament, emotional response, and habituation process in a horse. Our adapted behavioral test method proved successful in the evaluation of equine habituation processes with diverse behavior-related features. Our results suggested that generalization of new stimuli during habituation may differ depending on the horse’s temperament and tendencies in emotional response. Future adaptations of this work will aid in evidence of the differences in habituation process as a typical for Flightiness, Freeziness, Curiosity, or Timidity. Such findings will improve handler safety and lead to increased horse welfare during the habituation process. This is because the handler responses to the horse’s behavioral reactions toward habituation should interplay with the horse’s temperament.

## Figures and Tables

**Figure 1 animals-10-00921-f001:**
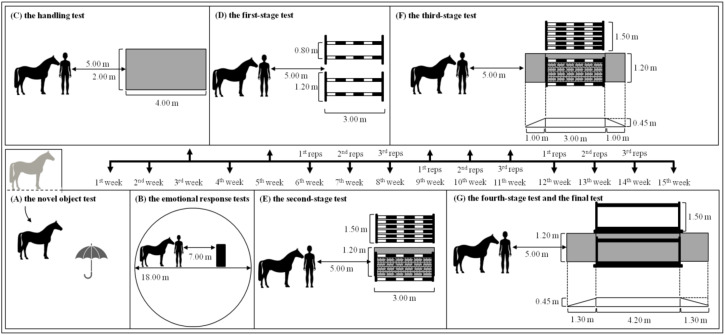
The timeline with marked stages of the experiment subsequent experiment procedures. (**A**) the novel object test; (**B**) the emotional response tests; (**C**) the handling test; (**D**) the first-stage test; (**E**) the second-stage test; (**F**) the third-stage test; (**G**) the fourth-stage test and the test session. Visual drawing is not to scale.

**Figure 2 animals-10-00921-f002:**
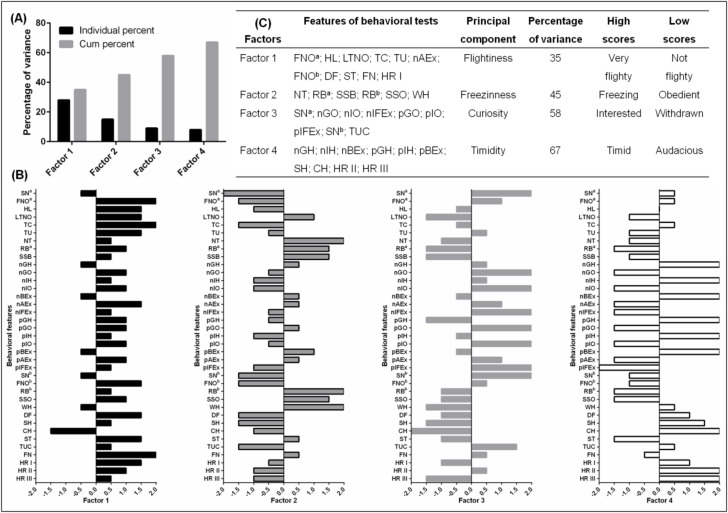
Classification of behavior-related features. (**A**) percent variance of four factors account for approximately 67% of the variance based on the principal component analysis (PCA) and factor analysis (FA); (**B**) behavior-related features loadings across each of the four identified factors. Items with an absolute value more than 1.5 describe the given principal component; (**C**) the ultimate features included in the analysis of relationships between horse’s temperament, emotional response, and habituation; (^a^) behaviors analyzed during evaluation of horse’s temperament; (^b^) behaviors analyzed during evaluation of horse’s habituation.

**Figure 3 animals-10-00921-f003:**
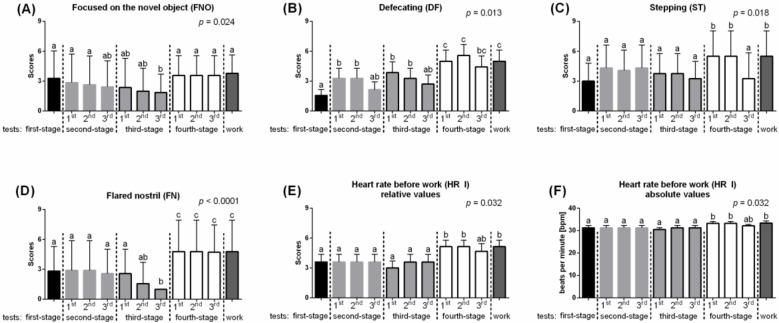
Behavioral differences between consecutive days of habituation in the “Flightiness” PCA. (**A**) Focused on the novel object (FNO); (**B**) Defecating (DF); (**C**) Stepping (ST); (**D**) Flared nostril; (**E**) Heart rate before work—relative values (HR I); (**F**) Heart rate before work—absolute values (HR I). The dashed lines marked a change of stage of the habituation tests. 1st, 2nd, 3rd marked the next repeat within the test. Different lowercase letters indicated significant differences between for *p* < 0.05. All values were scaled to 1–9 scale and reported as mean + SD.

**Figure 4 animals-10-00921-f004:**
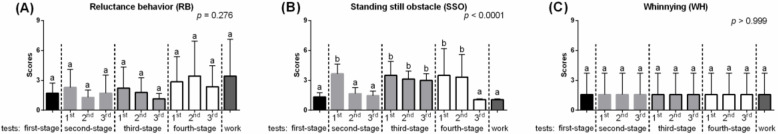
Behavioral differences between consecutive days of habituation in the “Freeziness” PCA. (**A**) Reluctance behavior (RB); (**B**) Standing still obstacle (SSO); (**C**) Whinnying (WH). The dashed lines marked a change of stage of the habituation tests. 1st, 2nd, 3rd marked the next repeat within the test. Different lowercase letters indicated significant differences between for *p* < 0.05. All values were scaled to 1–9 scale and reported as mean + SD.

**Figure 5 animals-10-00921-f005:**
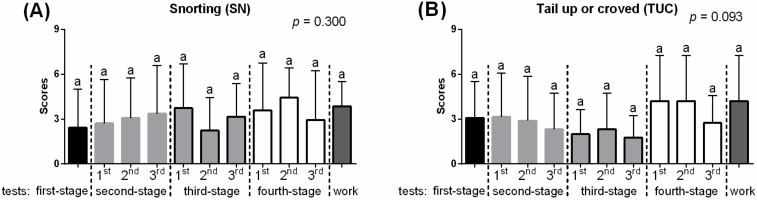
Behavioral differences between consecutive days of habituation in the “Curiosity” PCA. (**A**) Snorting (SN); (**B**) Tail up or coved (TUC). The dashed lines marked a change of stage of the habituation tests. 1st, 2nd, 3rd marked the next repeat within the test. Different lowercase letters indicated significant differences between for *p* < 0.05. All values were scaled to 1–9 scale and reported as mean + SD.

**Figure 6 animals-10-00921-f006:**
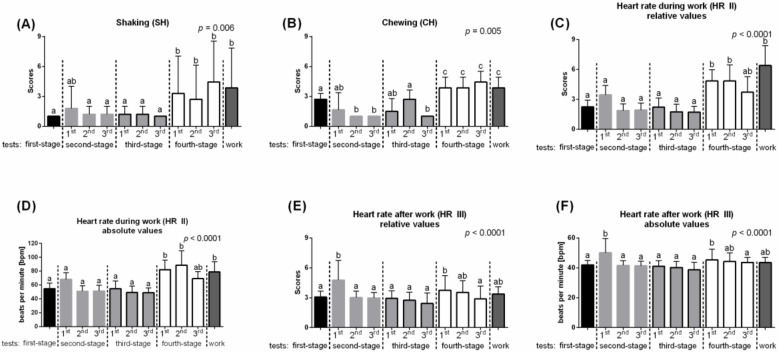
Behavioral differences between consecutive days of habituation in the “Timidity” PCA. (**A**) Shaking (SH); (**B**) Chewing (CH); (**C**) Heart rate during work—relative values (HR II); (**D**) Heart rate during work—absolute values (HR II); (**E**) Heart rate after work—relative values (HR III); (**F**) Heart rate after work—relative values (HR III). The dashed lines marked a change of stage of the habituation tests. 1st, 2nd, 3rd marked the next repeat within the test. Different lowercase letters indicated significant differences between for *p* < 0.05. All values were scaled to 1–9 scale and reported as mean + SD.

**Table 1 animals-10-00921-t001:** Ethogram of behaviors analyzed during evaluation of horse’s temperament.

Abreviation	Variable	Definition	Quantification
SN ^a^	Snorting	Forceful expulsion of air through the nostrils incidentally preceded by a raspy inhalation sound	frequency
FNO ^a^	Focused on the novel object	Ears, eyes, and head pointed in direction of novel object	percentage of total time
HL ^a^	Head low	Horse held its nose below its trunk line	percentage of total time
LTNO ^a^	Latency time	Latency time to touch the novel object for the first time	percentage of total time
TC ^a^	Trot and canter	Percentage of time trotting and cantering	percentage of total time
TU ^a^	Tail up	Percentage of time tail up when tail root is above horizontal line	percentage of total time
NT ^b^	Total needs	Total number of attempts needed to cross the bridge	frequency
RB ^b^	Reluctance behavior	Rearing, walking backwards, walking sideways, kicking, or head shaking while approaching the bridge	frequency
SSB ^b^	Standing still bridge	Percentage of time standing still in front of the bridge	percentage of total time

^a^ The Novel Object Test; ^b^ the Handling Test; the ethograms adapted from the studies of Visser et al. (2008) [25].

**Table 2 animals-10-00921-t002:** Ethogram of behaviors analyzed during evaluation of horse’s emotional response adapted from the studies of Schrimpf et al. (2020) [29].

Abreviation	Variable	Definition	Quantification
GH	Gaze human	Horse orients head towards human	frequency
GO	Gaze object	Horse orients head towards object	frequency
IH	Interaction human	Horse contacts physically with human	frequency
IO	Interaction object	Horse contacts physically with object	frequency
BEx	Behind handler	Percentage of time when horse’s is between the handler and close to the door, farthest from object	percentage of total time
AEx	Abreast to handler	Percentage of time when horse’s is next to handler between door and object	percentage of total time
IFEx	In front of handler	Percentage of time when horse’s is between the handler and the object	percentage of total time

**Table 3 animals-10-00921-t003:** Ethogram of behaviors analyzed during evaluation of horse’s habituation.

Abreviation	Variable	Definition	Quantification
SN ^a^	Snorting	Forceful expulsion of air through the nostrils incidentally preceded by a raspy inhalation sound	frequency
FNO ^a^	Focused on the novel object	Ears, eyes, and head pointed in direction of novel object	percentage of total time
RB ^a^	Reluctance behavior	Rearing, walking backwards, walking sideways, kicking or head shaking while approaching the obstacle	frequency
SSO ^a^	Standing still obstacle	Percentage of time standing still in front of the obstacle	percentage of total time
WH ^b^	Whinnying	The horse vocalizes	frequency
DF ^b^	Defecating	The horse drops feces	frequency
SH ^b^	Shaking	The trembling of the horse’s muscles is visible	frequency
CH ^b^	Chewing	The movements of the horse’s mouth and teeth are visible	frequency
ST ^b^	Stepping	Percentage of time when the horse isn’t standing stable on all four hoofs inside the obstacles but is stepping	percentage of total time
TUC ^b^	Tail up or croved	Percentage of time when the horse tail root is raised above horizontal line or is croved	percentage of total time
FN ^b^	Flared nostril	Percentage of time when the horse nostril is flared or flared with blowing	percentage of total time
HR I ^b^	Heart rate before work	Mean heart rate measured immediately before the work	beats per minute
HR II ^b^	Heart rate during work	Mean heart rate measured inside the obstacle or treadmill when all four hoofs were set down	beats per minute
HR III ^b^	Heart rate after work	Mean heart rate measured 30 s after the horse left the obstacle or treadmill and was stopped behind	beats per minute

^a^ Behaviors related to leading; the ethogram adapted from the studies of Yngvesson et al. (2016) and Stomp et al. (2018) [6,30]; ^b^ behaviors related to stress; the ethogram was from the studies of Yngvesson et al. (2016) and Padalino et al. (2018) [6,31].

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
