# Peer review of "Horse Behavior, Physiology and Emotions during Habituation to a Treadmill"

_animals, 2020, doi:10.3390/ani10060921_

Round 1

Reviewer 1 Report

Overall a great experiment, well design and very well performed. 

Some areas in the manuscript need significant improvement in the written english.

In my opinion, the introduction is lacking to provide evidence that the treadmill  issues that a horse and handler experience during treadmill tests mentioned in the manuscript are not seen in many referral places that use treadmill to further investigate poor performance problems and perform upper airway dynamic endoscopies. Please, if this is the case provide reference in the introduction and discussion. 

Line 3: Title: first (spelling mistake)

Line 17: I would use “has been investigated” rather than “was investigated”

Line 18: re-write the sentence of “behavioral observations,”, doesn’t make sense

Line 16-30: try to be consistent with English, past verbs and present. I recommend re-writing this section to make more readable

Line 32: I would add poor performance evaluation including dynamic endoscopy, etc

Line 38: How was the test done? Same person, different people assessing?

Line 50: add reference

Line 55: as mentioned above, I would add the use of the treadmill for assessment of poor performance, dynamic examination of the upper airway, etc.

Line 61: I would use “association” instead of “relations”

Line 77 to 79:  I recommend to split the sentence in 2 to make it flow better.

Line 107: I would use different word than “loaded” (this is more for transportation)

Line 125-126: it doesn’t make sense the first short sentence without continuation

Line 140: add and the “following” dimensions:

Line 163-164: I recommend to change the work “treatment” for test or any other alternative work, but I don’t think the horse were receiving a treatment. Please clarify.

Line 184: “all 4 feet were on the treadmill”

Line 192: was the quality of the videotape good enough to assess ethogram considering those camera don’t zoom in?

Line 237: screen instead of scree

Line 310-311: I would re-write. Difficult to understand

Line 352: heart rate doesn’Heret need to be with capital letter “H”

Line 372: “and” the behavioral tests

Line 448: need space between “that horses”

Line 449: human emotions

Author Response

We sincerely thank you for your response and the valuable comments on our manuscript. We are very pleased with your opinions that our experiment is well design and very well performed. We addressed all suggestions and pointed out our explanations below. We have a sincere hope that after all those improvements our work will be worth to be presented for the wider community.

1. In my opinion, the introduction is lacking to provide evidence that the treadmill issues that a horse and handler experience during treadmill tests mentioned in the manuscript are not seen in many referral places that use treadmill to further investigate poor performance problems and perform upper airway dynamic endoscopies. Please, if this is the case provide reference in the introduction and discussion.

We added in the introduction section this case provide reference however, it's not a major goal of our study, therefore this aspect was only mentioned.

  1. Line 3: Title: first (spelling mistake)

We changed the title following Reviewer 2 suggestion to "Horse behavior, physiology and emotions during habituation to a treadmill".

  1. Line 17: I would use “has been investigated” rather than “was investigated”

We agree and corrected this sentence.

  1. Line 18: re-write the sentence of “behavioral observations,”, doesn’t make sense

We re-wrote this sentence.

  1. Line 16-30: try to be consistent with English, past verbs and present. I recommend re-writing this section to make more readable

We re-wrote the Simple Summary.

  1. Line 32: I would add poor performance evaluation including dynamic endoscopy, etc

We added above.

  1. Line 38: How was the test done? Same person, different people assessing?

We added the information about staff in line 36.  

  1. Line 50: add reference

We add a citation and due to this change, we have also changed the order of the first three positions in the reference section and references to them throughout the manuscript.

  1. Line 55: as mentioned above, I would add the use of the treadmill for assessment of poor performance, dynamic examination of the upper airway, etc.

We added this assessment however as it was stated in point 1, it's not a major goal of our study, therefore this aspect was only mentioned.

  1. Line 61: I would use “association” instead of “relations”

We changed it.

  1. Line 77 to 79: I recommend to split the sentence in 2 to make it flow better.

We split the sentence.

  1. Line 107: I would use different word than “loaded” (this is more for transportation)

We replaced the word "loaded" with "entered".

  1. Line 125-126: it doesn’t make sense the first short sentence without continuation

We agree and removed the short sentence.

  1. Line 140: add and the “following” dimensions:

We added it.

  1. Line 163-164: I recommend to change the work “treatment” for test or any other alternative work, but I don’t think the horse were receiving a treatment. Please clarify.

We used the word “treatment” following Schrimpf et al. (2020) however, following your suggestion we changed it throughout the manuscript.

  1. Line 184: “all 4 feet were on the treadmill”

We changed it.

  1. Line 192: was the quality of the videotape good enough to assess ethogram considering those camera don’t zoom in?

Yes, the quality of the videotape was good. We used wide-angle digital camera and the digital zooming was possible at the level of records assessment.

  1. Line 237: screen instead of scree

We corrected this spelling mistake.

  1. Line 310-311: I would re-write. Difficult to understand

We re-wrote this sentence.

  1. Line 352: heart rate doesn’Heret need to be with capital letter “H”

We changed it.

  1. Line 372: “and” the behavioral tests

We added "and".

  1. Line 448: need space between “that horses”

We added space.

  1. Line 449: human emotions

We corrected it.

Reviewer 2 Report

This is an important paper on personality traits in horses. Habituation to novel situations is a very important part of horse training as the paper emphasise. 

Handling of horses with different personalities should, according to this paper, be handled according to their personalities. This is a fairly novel approach and very interesting. However the paper lacks data on handling strategies in connection to horse personalities so I would reccommend drawing these conclusions more carefully. Add the work by Malin Axel-Nilsson to the parts of the paper dealing with personalities and tests. 

This paper deals with learning in horses, especially the process of habituation. Please add, to the Introduction, a short paragraph on the fundamentals of the habituation process and how this is applied to horse training. Use the references more extensively and make sure all you write about learning is correct according to the current scientific status.

Title: I suggest the following title instead: "Horse behaviour, physiology and emotions during habituation to a treadmill." 

Abstract: Add the information abut heart rate measurements. 

L. 51 the word is "stereotypies" not "stereotypes".

L. 53-54 delete the part of the sentence "On the other hand, one of the ways to provide the horse with an adequate amount of movement is to work on the..." Working a horse on a treadmill can never be a way of fulfulling the horses' innate need for free movement. It also seems a very impractical way of exercising a horse, which is what this sentence implies. 

L. 54 how about just giving a brief description of what a treadmill is? Just to make it easier for the reader.

Materials & Methods

Please add much more details on the behaviour of the handlers. Was it always the same person? Exactly how were the horses handled during the habituation? This information is needed to be able to repeat the experiment. 

Also add more information on how you applied the HR-monitors.

L102-106 From the description of the housing of the horses it is likely that these horses are at least socially frustrated and they may also be frustrated due to too short eating times (roughage). This will have implications for their behaviour and affect the estimation of their personalities. It is important that you discuss potential effects of these factors on your methods and results. I think the results are still valid, as a large number of horses are housed this way but it it important to understand that if you take a horse from such a housing and put it in a more species correct type of housing the personality (e.g. fear responses) will differ.

L. 178 I would reccomend replacing the word "work" for "session".

L. 181-183 Please clarify in more detail how the negative reinforcement was applied. At exactly what behaviour in the horses did the lead rope tension release? And at exactly what behaviour did the horses get the positive reinforcement? Were the start and stop responses clear in the horses in all other situation of handling?

L. 186 The apple/carrot is the primary reinforcer and the voice command the secondary reinforcer. Were all the horses conditioned to the secondary reinforcer?

L. 189 An action that is added after an unwanted behaviour is per definition a positive punishment. Did you use positive punishment and in that case in what situations?

Table 3. Check spelling of whinnying. 

Results:

Please add the actual numbers of heart rate to give an estimate of the strenght of the horses' reaction (in comparison to baseline of course).

Figure 2. This is a complex figure and it would benefit from more explanations in the figure texts. Help the reader to remember what all abbreviations mean and what the units are on the axes. 

Discussion:

L. 418-419 I would like just a part of a sentence about the evolutionary aspects of the flight response in horses. You do write about this but I would just like you to clarify that this adaptation has been crucial for horses' survival during evolution and it is nothing domestication has fundamentally changed. However there may be breed and also individual differences (and it is those differences that you have seen).

Apart from this the early experiences of these 14 horses will have affected their behaviour as adults (and thereby their results in the personality tests). I would like to see just a sentence about how earlier experience might affect behaviour in the adults this in the discussion. You may not have any information about the early experience so just recognise the fact that this might have affected the horses' behaviour in the tests. 

L. 421-426 I would be very careful with the personality trait "stubborn". A freezing response due to fear is common in many animal species, including humans. But the word "stubborn" elicits an emotional (often negative) in human handlers. A negative emotional attitude from humans towards horses may impair horse welfare in a training situation. It seems from your results that the "stubborn" horses have a lower  heart rate than e.g. the timid horses, but it is unclear to me if that is what figure 2 actually says. This needs to be more clear and I would suggest choosing another lable for the stubborn horses. I am thinking of active and passive copers (e.g. in pig personality research), but I am unsure right now if this could be a parallell? 

L. 426 This sentence is unclear and needs to be re-written to facilitate understanding " In our research, the most stubborn horses were patients with various stimuli."

L. 457-478 Please clarify in detail what you mean with "classical handling methods during habituation" and "special handling methods during habituation, so-called alternative habituation". Also add references to this.  

Also discuss the ethical implications of your results and of the testing you exposed the horses and people to. 

I look forward to reading a revised version.

Author Response

Thank you very much for kind words about our paper and a substantial amount of time looking over the manuscript. We are very pleased with your opinion, that this is an important paper on personality traits in horses. We have a sincere hope that after thorough improvement following your comments our results will be worth to be presented for the wider community in the Animals.

  1. The paper lacks data on handling strategies in connection to horse personalities so I would recommend drawing these conclusions more carefully.

Thank you for this comment. In this paper we present the preliminary conclusions about the association between horses' temperament and behavioral habituation of horses to the work on a treadmill. We highlighted the lack of data on handling strategies in connection to horse personalities at the end of discussion section.

  1. Add the work by Malin Axel-Nilsson to the parts of the paper dealing with personalities and tests.

We added the work Axel-Nilsson, M.; Peetz-Nielsen, P.; Visser, E.K.; Nyman, S.; Blokhuis, H.J. Perceived relevance of selected behavioural traits in horses – A survey conducted in Sweden. Acta. Agr. Scand. A-An. 2015, 65, 23-32, doi: 10.1080/09064702.2015.1047791 as a sample of behavioral assessments in horses however, this work does not change the discussion of our results. We have also changed the order of the references from 34 in  reference section and throughout the manuscript.

  1. This paper deals with learning in horses, especially the process of habituation. Please add, to the Introduction, a short paragraph on the fundamentals of the habituation process and how this is applied to horse training. Use the references more extensively and make sure all you write about learning is correct according to the current scientific status.

We agree and tried to improve this aspect and add an appropriate new references in both introduction and discussion sections.

  1. Title: I suggest the following title instead: "Horse behaviour, physiology and emotions during habituation to a treadmill".

We changed the title according to your suggestion.

  1. Abstract: Add the information about heart rate measurements.

We added the information about heart rate measurements to abstract.

  1. L. 51 the word is "stereotypies" not "stereotypes".

We corrected it.

  1. L. 53-54 delete the part of the sentence "On the other hand, one of the ways to provide the horse with an adequate amount of movement is to work on the..." Working a horse on a treadmill can never be a way of fulfulling the horses' innate need for free movement. It also seems a very impractical way of exercising a horse, which is what this sentence implies.

We removed this sentence and replaced the reference with other use of the treadmill.

  1. L. 54 how about just giving a brief description of what a treadmill is? Just to make it easier for the reader.

We added short clarification of the bases of treadmill at the beginning of introduction section.

  1. Materials & Methods

Please add much more details on the behaviour of the handlers. Was it always the same person? Exactly how were the horses handled during the habituation? This information is needed to be able to repeat the experiment. Also add more information on how you applied the HR-monitors.

We added above information about handler, the behavior of handler and the HR-monitoring in the material and method section.

  1. L102-106 From the description of the housing of the horses it is likely that these horses are at least socially frustrated and they may also be frustrated due to too short eating times (roughage). This will have implications for their behaviour and affect the estimation of their personalities. It is important that you discuss potential effects of these factors on your methods and results. I think the results are still valid, as a large number of horses are housed this way but it important to understand that if you take a horse from such a housing and put it in a more species correct type of housing the personality (e.g. fear responses) will differ.

In the description of the horse's housing, there is no information about the length of the eating time. There is information that the horses were fed three times a day with a dose of oats and hay personalized to each horse to maintain optimal, healthy condition without obesity. Why the horses are exposed to stress due to insufficient fiber availability? Moreover, there is also information that horses daily spend a lot of time together on a paddock and have a daily portion of effective movement. We can conclude that both ways fulfill the horses' innate need for free movement. We can not agree with the statement that the horses are at least socially frustrated. Especially that horses care and experimental procedures were in accordance with the guidelines for the treatment of animals in behavioral research and teachings by the Association for the Study of Animal Behavior. We may agree that naive horses may express different behavior however, there are no technical possibilities to conduct so complicated experimental design in naive or wild horses. We conduct our experiment and discuss our results on horses' housing in a typical stable system, which, as you say, a large number of horses are housed this way. We may also agree that our type of housing is not natural however is as good as possible for horses kept in a stable system. There is no evidence of the social frustration of horses in this type of housing and we added this information to the material and methods section.

  1. L. 178 I would recommend replacing the word "work" for "session".

We may agree and replace the word "work" with work "session" in the context of final-first-full work on the treadmill throughout the manuscript, consequently.

12.L. 181-183 Please clarify in more detail how the negative reinforcement was applied. At exactly what behaviour in the horses did the lead rope tension release? And at exactly what behaviour did the horses get the positive reinforcement? Were the start and stop responses clear in the horses in all other situation of handling?

We added the clarification of the positive reinforcement and the positive punishment, we hope in sufficient detail.

  1. L. 186 The apple/carrot is the primary reinforcer and the voice command the secondary reinforcer. Were all the horses conditioned to the secondary reinforcer?

All horses were conditioned in the same way throughout the whole habituation process - to both the primary and the secondary reinforcer.

  1. L. 189 An action that is added after an unwanted behaviour is per definition a positive punishment. Did you use positive punishment and in that case in what situations?

Thank you very much for this valuable comment. We agree with yours definition and changed the negative reinforcement to the positive punishment throughout the manuscript, consequently.

  1. Table 3. Check spelling of whinnying.

We corrected "whinnying" throughout the whole manuscript also on the Figure 4.

  1. Results:

Please add the actual numbers of heart rate to give an estimate of the strenght of the horses' reaction (in comparison to baseline of course).

We can not change the scale of heart rate in a statistic model so we added the actual number of heart rate as an additional subfigure in the figure 3 (HR I) and figure 5 (HR II and HR III).

  1. Figure 2. This is a complex figure and it would benefit from more explanations in the figure texts. Help the reader to remember what all abbreviations mean and what the units are on the axes.

All subsection 3.1. The factors explaining behavioral variation is in fact the description of figure 2. Could you indicate which part could we better describe and in which place - in the subsection body or in the figure description to avoid the repetition of the data?

  1. Discussion:
  2. 418-419 I would like just a part of a sentence about the evolutionary aspects of the flight response in horses. You do write about this but I would just like you to clarify that this adaptation has been crucial for horses' survival during evolution and it is nothing domestication has fundamentally changed. However there may be breed and also individual differences (and it is those differences that you have seen).

Thank you very much for this important comment. We introduce this information as the first part of the discussion of individual trends. It is really important for our future work.

  1. Apart from this the early experiences of these 14 horses will have affected their behaviour as adults (and thereby their results in the personality tests). I would like to see just a sentence about how earlier experience might affect behaviour in the adults this in the discussion. You may not have any information about the early experience so just recognise the fact that this might have affected the horses' behaviour in the tests.

We sincerely thank you again for this important comment. We introduce this information in the first part of the discussion of individual trends.

  1. L. 421-426 I would be very careful with the personality trait "stubborn". A freezing response due to fear is common in many animal species, including humans. But the word "stubborn" elicits an emotional (often negative) in human handlers. A negative emotional attitude from humans towards horses may impair horse welfare in a training situation. It seems from your results that the "stubborn" horses have a lower heart rate than e.g. the timid horses, but it is unclear to me if that is what figure 2 actually says. This needs to be more clear and I would suggest choosing another lable for the stubborn horses. I am thinking of active and passive copers (e.g. in pig personality research), but I am unsure right now if this could be a parallell?

We may agree with your opinion an change the personality trait "stubborn" to "freeziness". We suspected this label is not connected with a negative emotional attitude from human and is more parallel than passive copers.

  1. L. 426 This sentence is unclear and needs to be re-written to facilitate understanding " In our research, the most stubborn horses were patients with various stimuli."

We agree and re-wrote this sentence.

  1. L. 457-478 Please clarify in detail what you mean with "classical handling methods during habituation" and "special handling methods during habituation, so-called alternative habituation". Also add references to this.

We added the clarification of the classical and so-called alternative habituation, we hope in sufficient detail.

Once again we sincerely thank you for your extremely valuable comments on our manuscript. We hope we were able to appropriately address it at least in part. We are looking forward to your second revision because your point of view sheds new light on our findings, thinking, and reasoning. We hope to focus on this view in the next research tasks.
